# Increasing the reach of evidence-based interventions for weight management and diabetes prevention among Medicaid patients: study protocol for a pilot Sequential Multiple Assignment Randomised Trial

Chelsey R Schlechter [1,2] Guilherme Del Fiol [3] Dusti R Jones,[1,2] Brian Orleans,[2] Bryan Gibson,[3] Inbal Nahum-Shani,[4] Ellen Maxfield,[5,6] Amy Locke [5,6] Ryan Cornia,[3] Richard Bradshaw,[3] Jennifer Wirth,[2] Shanna J Jaggers [2] Cho Y Lam,[1,2] David W Wetter[1,2]

For numbered affiliations see end of article.

**Correspondence to**
Dr Chelsey R Schlechter;
Chelsey.schlechter@hci.utah. edu

## ABSTRACT

**Introduction** Over 40% of US adults meet criteria for obesity, a major risk factor for chronic disease. Obesity disproportionately impacts populations that have been historically marginalised (eg, low socioeconomic status, rural, some racial/ethnic minority groups). Evidence-based interventions (EBIs) for weight management exist but reach less than 3% of eligible individuals. The aims of this pilot randomised controlled trial are to evaluate feasibility and acceptability of dissemination strategies designed to increase reach of EBIs for weight management.

**Methods and analysis** This study is a two-phase, Sequential Multiple Assignment Randomized Trial, conducted with 200 Medicaid patients. In phase 1, patients will be individually randomised to single text message (TM1) or multiple text messages (TM+). Phase 2 is based on treatment response. Patients who enrol in the EBI within 12 weeks of exposure to phase 1 (ie, responders) receive no further interventions. Patients in TM1 who do not enrol in the EBI within 12 weeks of exposure (ie, TM1 non-responders) will be randomised to either TM1-Continued (ie, no further TM) or TM1 & MAPS (ie, no further TM, up to 2 Motivation And Problem Solving (MAPS) navigation calls) over the next 12 weeks. Patients in TM+ who do not enrol in the EBI (ie, TM+ non-responders) will be randomised to either TM+Continued (ie, monthly text messages) or TM+ & MAPS (ie, monthly text messages, plus up to 2 MAPS calls) over the next 12 weeks. Descriptive statistics will be used to characterise feasibility (eg, proportion of patients eligible, contacted and enrolled in the trial) and acceptability (eg, participant opt-out, participant engagement with dissemination strategies, EBI reach (ie, the proportion of participants who enrol in EBI), adherence, effectiveness).

**Ethics and dissemination** Study protocol was approved by the University of Utah Institutional Review Board (#00139694). Results will be disseminated through study partners and peer-reviewed publications.

## STRENGTHS AND LIMITATIONS OF THIS STUDY

⇒ Uses an innovative Sequential Multiple Assignment Randomized Trial design to optimise adaptive interventions.
⇒ Potential reach and sustainability are enhanced by using dissemination strategy modalities (ie, cellphone-based voice and text) that are ubiquitous among adults in the USA, including among historically marginalised populations, and are already utilised by payers and healthcare systems.
⇒ Participant population was drawn from a single healthcare system in a single state.

**Trial registration number** clinicaltrials.gov; NCT05666323.

## INTRODUCTION

Approximately 40% of US adults meet criteria for obesity (BMI ≥30),[1] and this proportion is expected to rise to nearly 50% in the next decade.[2] Obesity is a leading risk factor for multiple chronic diseases, including at least 13 different cancers,[3–5] type 2 diabetes,[6 7] cardiovascular disease[8 9] and premature death.[10] Additionally, obesity plays a critical role in health inequities, disproportionately impacting low socioeconomic status (SES), rural and many racial/ethnic minority populations.[11–15] For example, individuals with lower levels of education have higher obesity rates compared with those with higher levels of education.[16] Recent studies have found obesity rates of 41% among rural US adults versus 32% among urban residents, and rising BMI among rural populations has

been a principal contributor to the obesity epidemic[17 18] over the last 30 years. Finally, obesity is more prevalent among Latino adults compared with non-Latino white adults (45.6% and 41.4%, respectively).[15]

## Evidence-based weight management and diabetes prevention programmes

Recommendations from the US Preventive Services Task Force and others highlight that adults with obesity should engage in comprehensive, multicomponent behavioural interventions addressing weight, diet and physical activity.[19–22] Components of these behavioural evidence-based interventions (EBIs) include improving diet and nutrition, increasing physical activity and behavioural strategies such as goal setting and self-monitoring.[23] The Diabetes Prevention Program (DPP) is among the most well-known EBIs for weight management, with substantial evidence for weight loss, increased physical activity, decreased daily energy intake and decreased incidence of type 2 diabetes.[24] The DPP can be delivered via in-person or digital modalities, and there are currently over 1800 Centers for Disease Control and Prevention (CDC) recognised DPP programmes across the USA.[25] Despite their effectiveness, DPPs are grossly underutilised,[26] reaching only 3% of eligible individuals.[27] Furthermore, there are substantial disparities in DPP access (14.6% of rural counties contain a DPP site vs 48.4% for urban counties)[28] and engagement among historically marginalised populations (eg, low SES, Latino).[26 29] Barriers to participation for in-person delivered DPPs among historically marginalised populations include the need to travel long distances, and the lack of transportation, financial resources, childcare and time. Fortunately, digital DPP delivery modalities such as the telephone, web-based and smartphones have yielded weight loss and other outcomes comparable to in-person DPPs, with greater engagement and lower attrition in digital programmes.[24 26 30–34] Consistent with those findings, the Community Preventive Services Task Force concluded that 'Technology-Supported Multicomponent Coaching or Counseling Interventions' are recommended for both weight loss and weight loss maintenance[20] and that technology-supported programmes may increase access among populations that have transportation issues, have unusual work schedules or are rural. Similarly, the National Academy of Medicine report on obesity treatment identified key research recommendations that included improving access, employing technology to increase reach and reducing health disparities.[21]

## Dissemination strategies to increase EBI reach

Dissemination strategies are designed to increase the reach of EBIs by increasing individual motivation and capability to access and engage with the interventions.[35 36] Many strategies to increase the reach of weight management EBIs rely on dissemination strategies at the point of care (eg, provider referral to EBI).[37–39] However, other potent strategies for increasing reach include proactive, population health management approaches that can remove the burden from clinical staff to offer the referral and can also reach out to all appropriate patients without the need for a clinical visit to trigger the referral offer. Population health management approaches have increased referrals to EBIs for weight management[27 37] and enrolment in EBIs among eligible participants.[40 41]

Digital and telehealth dissemination strategies such as text messaging and phone-based health coaching are promising approaches to improve dissemination of EBIs for weight management and have near ubiquitous reach given 97% of Americans own a cellphone, including 97% of individuals with income less than $30 000 per year, 94% of rural individuals and nearly 100% of Hispanic individuals.[42] Text messaging has been effective for improving numerous health behaviours (eg, physical activity, nutrition and tobacco cessation) and for improving attendance and compliance with behavioural interventions.[43–48] Similarly, personalised support to address barriers to care via patient navigation has been shown to improve completion rates for cancer screening, increase adherence to medication, improve compliance to treatment and follow-up visits, improve biomarkers for diabetes and increase enrolment in EBIs.[49–52] However, there is limited research on the effectiveness of these approaches for increasing reach of EBIs for weight management among historically marginalised populations.

## Adaptive interventions

Adaptive interventions contain a sequence of individually tailored decision rules that specify the intensity or type of intervention at critical points in the delivery of care.[53–55] Modifying intensity or type of intervention over time can improve outcomes if an individual is not responding or may reduce costs and burden if intensive treatment is not necessary.[56] For example, a stepped care model (a type of adaptive intervention) can conserve scarce resources and reduce burden on patients. Sequential Multiple Assignment Randomized Trials (SMART[57 58]) are a type of experimental design used to inform the dosage, type, delivery and timing of adaptive interventions.

## Aims and objectives

The overall aims of this pilot study are to evaluate the feasibility and acceptability of the dissemination strategies and study procedures. Feasibility will be measured by the proportion of people who were eligible, contacted and randomised in the trial. Acceptability will be assessed by rate of participant opt-out, participant engagement with dissemination strategies, EBI reach by randomisation condition, and EBI adherence and effectiveness among participants who enrol in the EBI.

## METHODS AND ANALYSIS

The protocol for this pilot study has been reviewed and approved by the University of Utah Institutional Review Board. As study interventions were deemed no more than

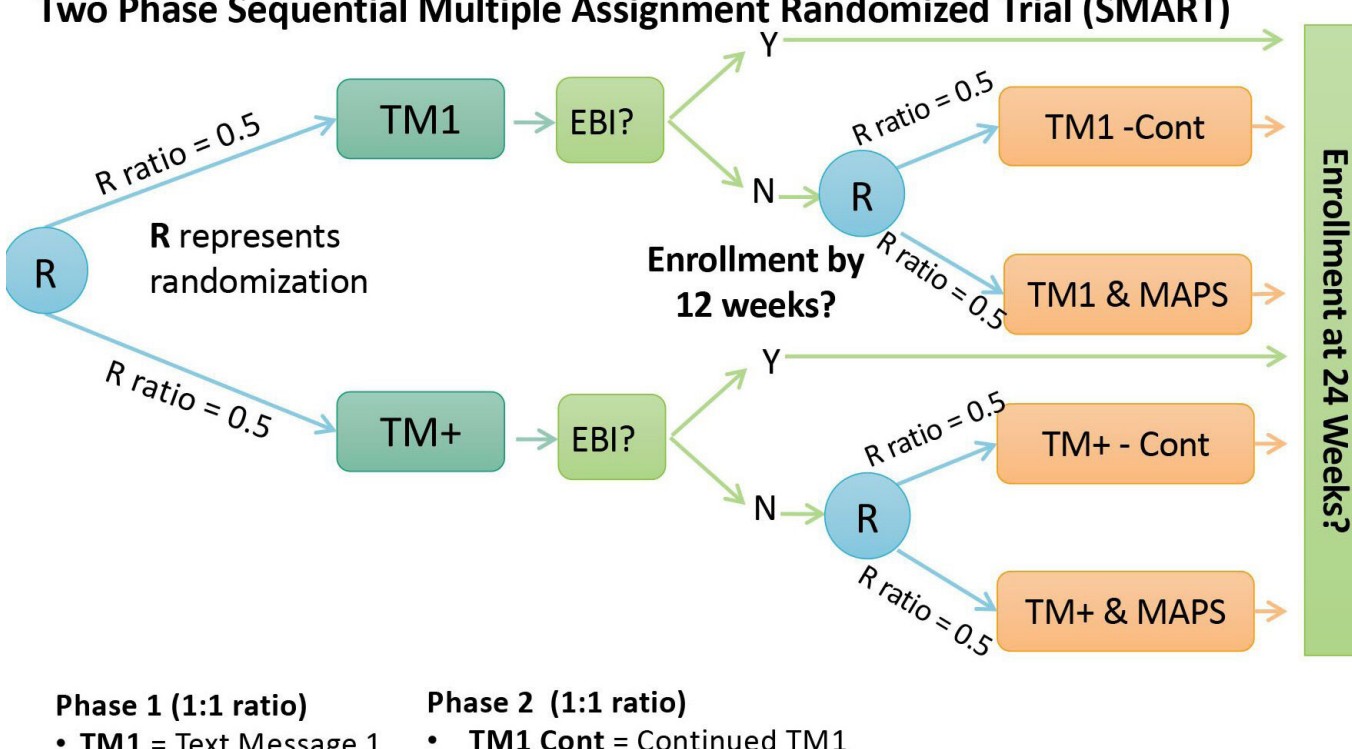

**Figure 1** Sequential Multiple Assignment Randomised Trial study design.

minimal risk, documentation of informed consent was waived for the project. All participants will be notified of the study and given the opportunity to opt-out of the project using a cover letter.

Data collection was planned to begin in March 2022 and end in May 2023. The trial is registered at clinical-trials.gov (NCT05666323). This manuscript adheres to the Standard Protocol Items: Recommendations for Interventional Trials.

### Settings and participants

Participants will be 200 adults (1) age ≥18; (2) BMI ≥30; (3) have had a primary care appointment within University of Utah Health (UHealth) within the past 12 months; (4) speak English or Spanish as their primary language; (5) have a current Utah address and (6) are insured through Medicaid by the University of Utah Health Plans. Exclusion criteria include (1) a diagnosis of type 1 or type 2 diabetes, (2) documented current pregnancy and (3) no phone number listed in the electronic health record (EHR).

### Study design

This trial is a two-phase SMART with 200 Medicaid patients from a single academic medical system (figure 1). In the first phase, participants will be individually randomised to receive a single text message (TM1) or multiple text messages (TM+) offering free enrolment in a digital EBI for weight management.

Phase 2 dissemination strategies are randomised based on prior dissemination strategy response. Participants in TM1 who do not opt out or enrol in the EBI by 12 weeks will be randomised to either TM1-Continued (no further TM) or TM1 & MAPS (no further TM and up to 2 Motivation And Problem Solving (MAPS) calls from a patient navigator). Participants in TM+ who do not opt out or enrol in the EBI by 12 weeks will be randomised to either TM+Continued (monthly text messages for the next 12 weeks) or TM+ & MAPS (monthly text messages for the next 12 weeks, plus up to 2 MAPS calls from a patient navigator). Descriptions of participant randomisation conditions are listed in table 1.

### Dissemination strategies
#### Text messaging

Participants will receive bidirectional Health Insurance Portability and Accountability Act compliant text messages that include a brief motivational message, the opportunity to enrol in the EBI, and the opportunity to opt out of receiving further dissemination strategies.

#### MAPS health coaching

Participants will receive up to two brief calls from health coaches trained in using MAPS, a coaching/counselling

**Table 1** Randomisation condition description

| | Randomisation condition | Description |
|---|---|---|
| **Phase 1** | | |
| | Single text message (TM1) | Participants receive a single text message that includes a motivational message, the EBI phone number, and a two-touch response that directly connects interested participants to EBI enrolment |
| | Multiple text message (TM+) | Participants receive messages once per week for the first 2 weeks, then 1 message every 3 weeks for the remaining 10 weeks (up to five texts total). The text messages include a motivational message, the EBI phone number and a simple two-touch response that directly connects interested participants to EBI enrolment |
| **Phase 2** | | |
| For TM1 non-responding participants only | Single text message continued (TM1-Cont) | Participants receive no additional text messages |
| | Single text message & Motivation And Problem Solving (MAPS) (TM1 & MAPS) | Participants receive no additional text messages and receive up to 2 MAPS counselling calls from a health coach |
| For TM+ non-responding participants only | Multiple text message continued (TM+-Cont) | Participants receive text messages every 4 weeks for the next 12 weeks (up to three text messages total) |
| | Multiple text message & MAPS (TM+ & MAPS) | Participants receive text messages every 4 weeks for the next 12 weeks (up to three text messages total) plus up to 2 MAPS counselling calls from a health coach |

EBI, evidence-based intervention.

approach designed to help participants address the larger context in which behaviour change occurs, identify and address barriers to change, and motivate participants towards a long-term goal of change. MAPS is a holistic, dynamic behavioural approach that integrates two empirically validated approaches (motivational interviewing[59 60] and problem-solving/social cognitive strategies[61–63]) into a coordinated intervention with demonstrated efficacy for addressing the entire process of considering behaviour change, attempting to change and maintaining change.[50 63–66] MAPS employs a 'wellness' approach that in addition to focusing on the target behaviour, provides practical advice and connections to additional services whenever possible for life events, stressors and concerns that impact the lives of rural and other historically marginalised individuals (eg, isolation, stress, depression, financial barriers, etc). MAPS is based on substantial evidence that motivation fluctuates frequently and rapidly and as such, is designed to fluidly respond to those changes.[67–70] MAPS health coaching calls will be delivered by two bilingual health coaches on the research team with at least 20 hours of training in MAPS delivery.

### Digital EBI for diabetes prevention and weight management

Participants who enrol in the EBI will receive either a digital DPP or digital weight management programme based on criteria for meeting pre-diabetes. Participants will be evaluated for risk of pre-diabetes using the American Diabetes Association (ADA) and CDC Prediabetes Risk Test[71] at the time of enrolling in the EBI. The test uses only self-report health trait data (eg, family history, height, weight, physical activity level, health history of hypertension and gestational diabetes) and does not require testing data (ie, haemoglobin A1C or cholesterol levels). Participants who meet criteria for pre-diabetes based on the risk test will be triaged into a digital DPP; participants who do not meet criteria will be triaged to a digital weight management programme. Both programmes will be delivered by EBI vendor, Incenta-HEALTH. Based on national data on eligibility for the DPP among individuals with obesity and data from the EBI provider, we estimate that ~50% of participants will be triaged to DPP and ~50% to weight management.[72]

The digital DPP offered to study participants is a CDC fully recognised programme. The digital weight management programme, for all participants who do not meet criteria for the DPP, uses the same general components as the DPP. Both programmes are designed to increase intake of fruits, vegetables, lean protein and complex carbohydrates while monitoring portion size. Physical activity recommendations include cardiovascular and strength training, with a goal of approximately 30 min 5–6 days/week. Both programmes are offered in English and Spanish, based on social cognitive theory, and provide personalised health coaching. Both include a website, a Bluetooth wireless scale for weight monitoring, daily email/text message support, easy to follow video exercise plans, customised meal plans, grocery lists, incorporate skill building and incorporate social accountability. Participants have unlimited access to health coaches, who have at least a Bachelor's degree,

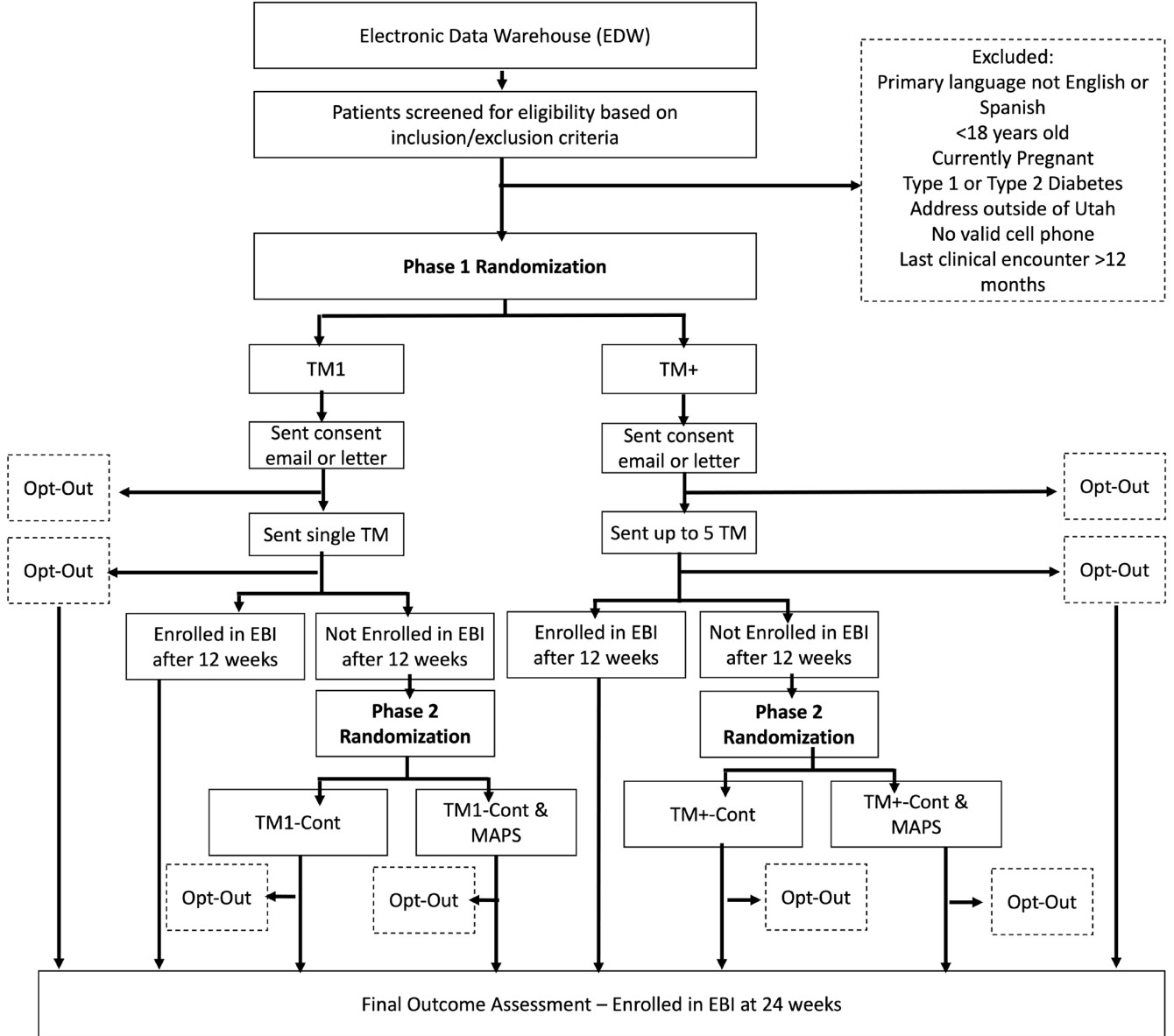

**Figure 2** Patient flow diagram.

along with certifications, in personal training, nutrition or wellness.

## Recruitment

The flow of participants through the study is described in figure 2. Participants who meet inclusion criteria will be identified from the UHealth Electronic Data Warehouse (EDW). Participant demographic, preferred language and relevant health data, as well as participant contact information (ie, phone number, address, email address), will be extracted from the EDW for all eligible participants. Participants who meet exclusionary criteria will be removed from the eligible population. We will use quota sampling from the eligible population to achieve a stratified random sample aimed to offer balance in the marginal distributions of key participant demographic characteristics. These

margins include (1) male/female sex (1:1 ratio), (2) urban/rural according to participant zip code and the 2010 Rural Urban Commuting Area codes (≥4 defined as rural) (1:1 ratio) and (3) Latino with English as preferred language, Latino with Spanish as preferred language, and non-Latino with English as preferred language (1:1:3 ratio).

Sampled patients randomised to TM1 or TM+ will be sent an email informing them that they will be receiving communications from UHealth about the availability of a digital EBI for weight management. The email will provide patients with information about how they can opt out if they do not wish to receive these communications. Patients who do not have an email in the EHR will be sent the opt-out information via mail. Patients who opt out of the study receive no further interventions.

## Randomisation and blinding

In phase 1 (figure 1), participants will be randomly assigned to TM1 or TM+ in a 1:1 ratio. In phase 2 (figure 1), participants who opt out or who enrol in the EBI will receive no further dissemination strategies and will not be rerandomised. Participants in TM1 did not opt out and who did not enrol in the EBI will be randomised to TM1-Cont or TM1 & MAPS at a ratio of 1:1. Participants in TM+ who did not opt out and did not enrol in the EBI will be randomised to TM+-Cont or TM+ & MAPS at a ratio of 1:1. All randomisation sequences will be generated via SAS 9.4 to consist of random permuted blocks with random block sizes, thus ensuring balance in the allocation over time.

## Data collection and outcomes

Participant demographic and baseline health data (eg, age, gender, BMI, pregnancy status, medical codes for type 1 and type 2 diabetes, zip code, preferred language, race/ethnicity) will be obtained from the UHealth EDW.

### Feasibility outcomes

Feasibility will be assessed using the proportion of participants who were eligible, contacted and enrolled in the trial.

### Acceptability outcomes

Acceptability will be assessed using participant opt-out, participant engagement with dissemination strategies, EBI reach and EBI effectiveness. Participant opt-out is assessed by the absolute number and proportion of eligible participants who opt out during phases 1 and 2. Participant engagement will be assessed as (1) the number and proportion of participants who respond to text messages with a response other than opt-out, (2) the number and proportion of eligible participants who talk to a MAPS health coach, (3) the number and proportion of eligible participants accept connection to the EBI and (4) the number and proportion of participants who talk to the EBI personnel. EBI reach is assessed by the number and proportion of eligible participants who officially enrol in the EBI. Enrolment is defined in accordance with the CDC definition for recognised virtual programmes, which consists of the participant setting a password for the programme. Enrolment will be obtained from the EBI vendor (IncentaHEALTH). EBI adherence and effectiveness are measured by participant adherence to the EBI components, percentage of body weight lost and change in BMI and will be collected for 12 weeks following EBI enrolment. Adherence and effectiveness data will be obtained from the EBI vendor (IncentaHEALTH).

## Patient and public involvement

This study was developed and implemented in collaboration with the University of Utah Wellness and Integrative Health Program, which supports population health initiatives through integration of wellness services into patient care and health plan offerings. The programme currently manages lifestyle behaviour change programmes (including the DPP) for UHealth patients and University of Utah faculty, staff and students. The dissemination strategies, the feasibility and acceptability outcomes, and study procedures were developed in collaboration with the University of Utah Wellness and Integrative Health Program and investigators at the University of Utah. Patients were not directly involved in the development of this pilot study.

## Ethics and dissemination

All procedures performed in studies involving human participants will be conducted in accordance with the ethical standards of the institutional and/or national research committee and with the 1964 Helsinki declaration and its later amendments or comparable ethical standards. The protocol for this study was approved by the University of Utah Institutional Review Board (00139694). Materials used to conduct the study are not currently publicly available. Materials may be requested by emailing the corresponding author. Study results will be dissemination via peer-reviewed publications and manuscripts, as well as to the health system and community partners via lay reports and presentations.

## Data analysis plan

Our primary study aim is to evaluate the acceptability and feasibility of dissemination strategies to increase of EBIs for weight management.

### Primary analyses phase 1

The analysis of acceptability and feasibility at 12 weeks will include all participants enrolled in the study. We will use descriptive statistics (mean, SD, CI) to evaluate each indicator of feasibility and acceptability by dissemination strategies (ie, TM1, TM+).

### Primary analyses phase 2

The analysis of feasibility and acceptability at 24 weeks will be conducted using only data from those participants who did not opt out or enrol in the EBIs by 12 weeks (ie, non-responders). We will use descriptive statistics to evaluate each indicator of feasibility and acceptability by dissemination strategies (ie, TM1-Cont, TM1 & MAPS, TM+-Cont, TM+ & MAPS).

## Power analysis

The sample size (n=200) for the current project was determined based on feasibility of sample size recruitment from a single Medicaid provider and completing project procedures in the study timeframe. As such, power calculations were not conducted for this pilot trial.

## Strengths and limitations

The proposed pilot study uses an innovative SMART design to optimise adaptive interventions to increase reach of already existing EBIs for weight management among historically marginalised populations. Potential reach and sustainability are enhanced by using dissemination strategy modalities (ie, cellphone-based voice

and text) that are ubiquitous among adults in the USA, including among historically marginalised populations, and are already utilised by payers and healthcare systems. The participant population was drawn from a single healthcare system in a single state which may limit generalisability of results. However, the proposed study population will represent a diverse sample based on sex, ethnicity, primary language and rurality.

## CONCLUSION

The proposed pilot project is a two-phase SMART with the objective of evaluating acceptability and feasibility of dissemination strategies designed to increase the reach of digital EBIs for weight management among Medicaid patients. Accomplishing the aims of the proposed study will provide critical data regarding the feasibility and acceptability of pragmatic and scalable dissemination strategies designed to increase the uptake of digital EBIs for obesity among historically marginalised populations and inform the conduct of a future, fully powered randomised trial.

**Author affiliations**
¹Department of Population Health Sciences, University of Utah, Salt Lake City, Utah, USA
²Huntsman Cancer Institute, University of Utah, Salt Lake City, Utah, USA
³Department of Biomedical Informatics, University of Utah, Salt Lake City, Utah, USA
⁴Institute for Social Research, University of Michigan, Ann Arbor, Michigan, USA
⁵Osher Center for Integrative Health, University of Utah, Salt Lake City, Utah, USA
⁶Department of Family & Preventive Medicine, University of Utah, Salt Lake City, Utah, USA

**Contributors**  DWW acquired financial support for the project. CRS, GDF, BO, BG, EM, AL, JW, CYL, RB, RC, SJJ, INS and DWW contributed to study conceptualisation. All authors contributed to the development or design of the methodology, dissemination strategies, study procedures, study outcomes or analytic plan. CRS and DRJ drafted the original manuscript. All authors reviewed, edited and approved the final manuscript.

**Funding**  This work was supported through the Ben B and Iris M Margolis Foundation, the National Center for Advancing Translational Sciences of the National Institutes of Health (UL1TR002538), the National Cancer Institute (P30CA042014) and the Huntsman Cancer Foundation.

**Competing interests**  None declared.

**Patient and public involvement**  Patients and/or the public were involved in the design, or conduct, or reporting, or dissemination plans of this research. Refer to the Methods section for further details.

**Patient consent for publication**  Not applicable.

**Provenance and peer review**  Not commissioned; externally peer reviewed.

**ORCID iDs**
Chelsey R Schlechter http://orcid.org/0000-0002-8355-6316
Guilherme Del Fiol http://orcid.org/0000-0001-9954-6799
Amy Locke http://orcid.org/0000-0002-6127-5361
Shanna J Jaggers http://orcid.org/0009-0006-7191-9489

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
