## [Reviewer comments · BMJ Open]

ARTICLE DETAILS

TITLE (PROVISIONAL)	Increasing the Reach of Evidence-Based Interventions for Weight Management and Diabetes Prevention Among Medicaid Patients: Study Protocol for a Pilot Sequential Multiple Assignment Randomized Trial.
AUTHORS	Schlechter, Chelsey; Del Fiol, Guilherme; Jones, Dusti; Orleans, Brian; Gibson, Bryan; Nahum-Shani, Inbal; Maxfield, Ellen; Locke, Amy; Cornia, Ryan; Bradshaw, Richard; Wirth, Jennifer; Jagggers, Shanna; Lam, Cho; Wetter, David

VERSION 1 – REVIEW

REVIEWER	Hesketh, Katie Liverpool John Moores University
REVIEW RETURNED	07-Jul-2023

GENERAL COMMENTS	An interesting concept to increase participation into EBIs - looking forward to seeing the results. A few suggestions if it is possible/feasible: 1) Collect follow-up data from all those enrolled on EBI 2) Possibility of accessing medical records for those who do not opt in to an EBI to get comparable data? 3) Interviews with participants to assess acceptability of text messages/counselling meetings for patient perspective to support the findings
---

REVIEWER	O'Neal, LaToya University of Florida, Family, Youth and Community Sciences
REVIEW RETURNED	13-Jul-2023

GENERAL COMMENTS	Overall, well written manuscript. Authors could expand on limitations and strengths in the manuscript. Minor grammatical errors should be addressed during editing process.
---

VERSION 1 – AUTHOR RESPONSE

Reviewer: 1

An interesting concept to increase participation into EBIs - looking forward to seeing the results.
Thank you!

A few suggestions if it is possible/feasible:

1) Collect follow-up data from all those enrolled on EBI. We thank the reviewer for this comment. We will collect follow-up data from all those enrolled in the EBI regardless of EBI type, and have further clarified this in the manuscript.

2) Possibility of accessing medical records for those who do not opt in to an EBI to get comparable data?

EW thank the reviewer for pointing out this clarification for us. We will have access to the EHR for all patients invited to participate in the study. This has been further clarified in the manuscript.

3) Interviews with participants to assess acceptability of text messages/counselling meetings for patient perspective to support the findings.

We completely agree with the reviewer that patient interviews would allow us to further explore qualitative data for acceptability of the messages and would add an additional, important perspective. However, due to limitations in project funding we are unable to add this piece of data collection.

Reviewer: 2

Overall, well written manuscript. Authors could expand on limitations and strengths in the manuscript. Minor grammatical errors should be addressed during editing process.

We thank the reviewer for their comments. We have expanded on limitations and strengths in the manuscript, and have thoroughly reviewed for grammatical errors.